# Regularization for Deep Learning: A Taxonomy

## Abstract

Regularization is one of the crucial ingredients of deep learning, yet the term *regularization* has various definitions, and regularization methods are often studied separately from each other. In our work we present a novel, systematic, unifying taxonomy to categorize existing methods. We distinguish methods that affect data, network architectures, error terms, regularization terms, and optimization procedures. We identify the atomic building blocks of existing methods, and decouple the assumptions they enforce from the mathematical tools they rely on. We do not provide all details about the listed methods; instead, we present an overview of how the methods can be sorted into meaningful categories and sub-categories. This helps revealing links and fundamental similarities between them. Finally, we include practical recommendations both for users and for developers of new regularization methods.

## 1 Introduction

Regularization is one of the key elements of machine learning, particularly of deep learning (Goodfellow et al., 2016), allowing to generalize well to unseen data even when training on a finite training set or with an imperfect optimization procedure. In the traditional sense of optimization and also in older neural networks literature, the term "regularization" is reserved solely for a penalty term in the loss function (Bishop, 1995a). Recently, the term has adopted a broader meaning: Goodfellow et al. (2016, Chap. 5) loosely define it as *"any modification we make to a learning algorithm that is intended to reduce its test error but not its training error"*. We find this definition slightly restrictive and present our working definition of regularization, since many techniques considered as regularization do reduce the training error (e.g. weight decay in AlexNet (Krizhevsky et al., 2012)).

**Definition 1. Regularization** is any supplementary technique that aims at making the model generalize better, i.e. produce better results on the test set.

This can include various properties of the loss function, the loss optimization algorithm, or other techniques. Note that this definition is more in line with machine learning literature than with inverse problems literature, the latter using a more restrictive definition.

In this work, we create a novel, systematic, unifying taxonomy of regularization methods for deep learning. We analyze existing methods and identify their atomic building blocks. This leads to decoupling of two important concepts: Which assumptions the methods rely on (and try to enforce), and which mathematical and algorithmic tools they use. In turn, this enables better understanding of existing methods and speeds up development of new ones: The researchers can focus either on finding new, better ways of enforcing existing assumptions, or focus on discovery of new assumptions that can be enforced in some existing way.

Before we proceed to the presentation of our taxonomy, we revisit some basic machine learning theory in Section 2. This will provide a justification of the top level of the taxonomy. In Sections 3–7, we continue with a finer division of the individual classes of the regularization techniques, aiming at separating as many clearly separable concepts as possible and isolating atomic building blocks of individual methods. Finally, in Section 8 we present our

practical recommendations for using existing methods and designing new methods. We are aware that the many research works discussed in this taxonomy cannot be summarized in a single sentence. For the sake of structuring the multitude of papers, we decided to merely describe a certain subset of their properties according to the focus of our taxonomy.

## 2 THEORETICAL FRAMEWORK

The central task of our interest is model fitting: finding a function $f$ that can well approximate a desired mapping from inputs $x$ to desired outputs $f(x)$. A given input $x$ can have an associated target $t$ which dictates the desired output $f(x)$ directly (or in some applications indirectly (Ulyanov et al., 2016; Johnson et al., 2016)). A typical example of having available targets $t$ is supervised learning. Data samples $(x, t)$ then follow a ground truth probability distribution $P$.

In many applications, neural networks have proven to be a good family of functions to choose $f$ from. A neural network is a function $f_w : x \mapsto y$ with trainable weights $w \in W$. *Training* the network means finding a weight configuration $w^*$, which is a result of performing a minimization procedure of a *loss function* $\mathcal{L} : W \to \mathbb{R}$ as follows:

$$w^* = \text{minimize } \mathcal{L}(w). \tag{1}$$

Usually the loss function takes the form of *expected risk*:

$$\mathcal{L} = \mathbb{E}_{(x,t) \sim P}\Big[E\big(f_w(x), t\big) + R(\ldots)\Big], \tag{2}$$

where we identify two parts, an *error function* $E$ and a *regularization term* $R$. The error function depends on the targets and assigns a penalty to model predictions according to their *consistency* with the targets. The regularization term assigns a penalty to the model based on other criteria. It may depend on anything except the targets, for example on the weights (see Section 6).

The expected risk cannot be minimized directly since the data distribution $P$ is unknown. Instead, a *training set* $\mathcal{D}$ sampled from the distribution is given. The minimization of the expected risk can be then approximated by (approximately) minimizing the *empirical risk* $\hat{\mathcal{L}}$:

$$\underset{w}{\text{minimize}} \frac{1}{|\mathcal{D}|} \sum_{(x_i, t_i) \in \mathcal{D}} E\big(f_w(x_i), t_i\big) + R(\ldots) \tag{3}$$

where $(x_i, t_i)$ are samples from $\mathcal{D}$.

Now we have the minimal background to formalize the division of regularization methods into a systematic taxonomy. In the minimization of the empirical risk, Eq. (3), we can identify the following elements that are responsible for the value of the learned weights, and thus can contribute to regularization:

- $\mathcal{D}$: The training set, discussed in Section 3
- $f$: The selected model family, discussed in Section 4
- $E$: The error function, briefly discussed in Section 5
- $R$: The regularization term, discussed in Section 6
- The optimization procedure itself, discussed in Section 7

Ambiguity regarding the splitting of methods into these categories and their subcategories is discussed in Appendix A using notation from Section 3.

## 3 REGULARIZATION VIA DATA

The quality of a trained model depends largely on the training data. Apart from acquisition/selection of appropriate training data, it is possible to employ regularization via data. This is done by applying some transformation to the training set $\mathcal{D}$, resulting in a new

set $\mathcal{D}_R$. Some transformations perform feature extraction or pre-processing, modifying the feature space or the distribution of the data to some representation simplifying the learning task. Other methods allow generating new samples to create a larger, possibly infinite, *augmented* dataset. These two principles are somewhat independent and may be combined. The goal of regularization via data is either one of them, or the other, or both. They both rely on *transformations with (stochastic) parameters*:

**Definition 2. Transformation with stochastic parameters** is a function $\tau_\theta$ with parameters $\theta$ which follow some probability distribution.

In this context we consider $\tau_\theta$ which can operate on network inputs, activations in hidden layers, or targets. An example of a transformation with stochastic parameters is the corruption of inputs by Gaussian noise (Bishop, 1995b; An, 1996):

$$\tau_\theta(x) = x + \theta, \quad \theta \sim \mathcal{N}(\mathbf{0}, \mathbf{\Sigma}). \tag{4}$$

The stochasticity of the transformation parameters is responsible for generating new samples, i.e. *data augmentation*. Note that the term *data augmentation* often refers specifically to transformations of inputs or hidden activations, but here we also list transformations of targets for completeness. The exception to the stochasticity is when $\theta$ follows a delta distribution, in which case the transformation parameters become deterministic and the dataset size is *not* augmented.

We can categorize the data-based methods according to the properties of the used transformation and of the distribution of its parameters. We identify the following criteria for categorization (some of them later serve as columns in Tables 1–2):

**Stochasticity of the transformation parameters $\theta$**

- Deterministic parameters: Parameters $\theta$ follow a delta distribution, size of the dataset remains unchanged
- Stochastic parameters: Allow generation of a larger, possibly infinite, dataset. Various strategies for sampling of $\theta$ exist:
  - Random: Draw a random $\theta$ from the specified distribution
  - Adaptive: Value of $\theta$ is the result of an optimization procedure, usually with the objective of maximizing the network error on the transformed sample (such "challenging" sample is considered to be the most informative one at current training stage), or minimizing the difference between the network prediction and a predefined fake target $t'$
    * Constrained optimization: $\theta$ found by maximizing error under hard constraints (support of the distribution of $\theta$ controls the strongest allowed transformation)
    * Unconstrained optimization: $\theta$ found by maximizing modified error function, using the distribution of $\theta$ as weighting (proposed herein for completeness, not yet tested)
    * Stochastic: $\theta$ found by taking a fixed number of samples of $\theta$ and using the one yielding the highest error

**Effect on the data representation**

- Representation-preserving transformations: Preserve the feature space and attempt to preserve the data distribution
- Representation-modifying transformations: Map the data to a different representation (different distribution or even new feature space) that may disentangle the underlying factors of the original representation and make the learning problem easier

**Transformation space**

- Input: Transformation is applied to $x$

- Hidden-feature space: Transformation is applied to some deep-layer representation of samples (this also uses parts of $f$ and $w$ to map the input into the hidden-feature space; such transformations act inside the network $f_w$ and thus can be considered part of the architecture, additionally fitting Section 4)
- Target: Transformation is applied to $t$ (can only be used during the training phase since labels are not shown to the model at test time)

**Universality**

- Generic: Applicable to all data domains
- Domain-specific: Specific (handcrafted) for the problem at hand, for example image rotations

**Dependence of the distribution of $\theta$**

- $p(\theta)$: distribution of $\theta$ is the same for all samples
- $p(\theta|t)$: distribution of $\theta$ can be different for each target (class)
- $p(\theta|t')$: distribution of $\theta$ depends on desired (fake) target $t'$
- $p(\theta|x)$: distribution of $\theta$ can be different for each input vector (with implicit dependence on $f$ and $w$ if the transformation is in hidden-feature space)
- $p(\theta|\mathcal{D})$: distribution of $\theta$ depends on the whole training dataset
- $p(\theta|\mathbf{x})$: distribution of $\theta$ depends on a batch of training inputs (for example (parts of) the current mini-batch, or also previous mini-batches)
- $p(\theta|\text{time})$: distribution of $\theta$ depends on time (current training iteration)
- $p(\theta|\pi)$: distribution of $\theta$ depends on some trainable parameters $\pi$ subject to loss minimization (i.e. the parameters $\pi$ evolve during training along with the network weights $w$)
- Combinations of the above, e.g. $p(\theta|x,t)$, $p(\theta|x,\pi)$, $p(\theta|x,t')$, $p(\theta|x,\mathcal{D})$, $p(\theta|t,\mathcal{D})$, $p(\theta|x,t,\mathcal{D})$

**Phase**

- Training: Transformation of training samples
- Test: Transformation of test samples, for example multiple augmented variants of a sample are classified and the result is aggregated over them

A review of existing methods that use generic transformations can be found in Table 1. Dropout in its original form (Hinton et al., 2012; Srivastava et al., 2014) is one of the most popular methods from the generic group, but also several variants of Dropout have been proposed that provide additional theoretical motivation and improved empirical results (Standout (Ba and Frey, 2013), Random dropout probability (Bouthillier et al., 2015), Bayesian dropout (Maeda, 2014), Test-time dropout (Gal and Ghahramani, 2016)).

Table 2 contains a list of some domain-specific methods focused especially on the image domain. Here the most used method is rigid and elastic image deformation.

**Target-preserving data augmentation**   In the following, we discuss an important group of methods: *target-preserving data augmentation*. These methods use **stochastic** transformations in **input** and **hidden-feature** spaces, while preserving the original target $t$. As can be seen in the respective two columns in Tables 1–2, most of the listed methods have exactly these properties. These methods transform the training set to a distribution $Q$, which is used for training instead. In other words, the training samples $(x_i, t_i) \in \mathcal{D}$ are replaced in the empirical risk loss function (Eq. (3)) by augmented training samples $(\tau_\theta(x_i), t_i) \sim Q$. By randomly sampling the transformation parameters $\theta$ and thus creating many new samples $(\tau_\theta(x_i), t_i)$ from each original training sample $(x_i, t_i)$, data augmentation attempts to

| Method | Dependence | Transformation space | Stochasticity ($\theta$ sampling) | Phase |
|---|---|---|---|---|
| Gaussian noise on input (Bishop, 1995a; An, 1996) | $p(\theta)$ | Input | Random | Training |
| Gaussian noise on hidden units (DeVries and Taylor, 2017) | $p(\theta)$ | Hidden features | Random | Training |
| Dropout (Hinton et al., 2012; Srivastava et al., 2014) | $p(\theta)$ | Input and hidden features | Random | Training |
| Random dropout probability (Bouthillier et al., 2015, Sec. 4) | $p(\theta)$ | Input and hidden features | Random | Training |
| Curriculum dropout (Morerio et al., 2017) | $p(\theta\|\text{time})$ | Input and hidden features | Random | Training |
| Bayesian dropout (Maeda, 2014) | $p(\theta\|\pi)$ | Input and hidden features | Random | Training |
| Standout (adaptive dropout) (Ba and Frey, 2013) | $p(\theta\|x, \pi)$ | Input and hidden features | Random | Training |
| "Projection" of dropout noise into input space (Bouthillier et al., 2015, Sec. 3) | $p(\theta\|x, f, w)$ | Input Uses auxiliary $\tau$ in hidden-feature space. | Random | Training |
| Approximation of Gaussian process by test-time dropout (Gal and Ghahramani, 2016) | $p(\theta)$ | Input and hidden features | Random | Test |
| Stochastic depth (Huang et al., 2016b) | $p(\theta)$ | Hidden features | Random | Training |
| Noisy activation functions (Nair and Hinton, 2010; Xu et al., 2015; Gülçehre et al., 2016a) | $p(\theta\|x)$ | Hidden features | Random | Training |
| Training with adversarial examples (Szegedy et al., 2014) | $p(\theta\|x, t')$ | Input | Adaptive Constrained | Training |
| Network fooling (adversarial examples) (Szegedy et al., 2014) *(Not for regularization)* | $p(\theta\|x, t')$ | Input | Adaptive Constrained | Test |
| Synthetic minority oversampling in hidden-feature space (Wong et al., 2016) | $p(\theta\|x, t, \mathcal{D})$ | Hidden features | Random | Training |
| Inter- and extrapolation in hidden-feature space (DeVries and Taylor, 2017) | $p(\theta\|x, t, \mathcal{D})$ | Hidden features | Random | Training |
| Batch normalization (Ioffe and Szegedy, 2015), Ghost batch normalization (Hoffer et al., 2017) | $p(\theta\|\mathbf{x})$ | Hidden features | Deterministic | Training and test |
| Layer normalization (Ba et al., 2016) | $p(\theta\|x)$ | Hidden features | Deterministic | Training and test |
| Annealed noise on targets (Wang and Principe, 1999) | $p(\theta\|\text{time})$ | Target | Random | Training |
| Label smoothing (Szegedy et al., 2016, Sec. 7; Goodfellow et al., 2016, Chap. 7) | $p(\theta)$ | Target | Deterministic | Training |
| Model compression (mimic models, distilled models) (Bucilă et al., 2006; Ba and Caruana, 2014; Hinton et al., 2015) | $p(\theta\|x, \mathcal{D})$ | Target | Deterministic | Training |

Table 1: Existing *generic* data-based methods classified according to our taxonomy. Table columns are described in Section 3.

| Method | Dependence | Transformation space | Stochasticity ($\theta$ sampling) | Phase |
|---|---|---|---|---|
| Rigid and elastic image transformation (Baird, 1990; Yaegger et al., 1996; Simard et al., 2003; Ciresan et al., 2010) | $p(\theta)$ | Input | Random | Training |
| Test-time image transformations (Simonyan and Zisserman, 2015; Dieleman et al., 2015) | $p(\theta)$ | Input | Random | Test |
| Sound transformations (Salamon and Bello, 2017) | $p(\theta)$ | Input | Random | Training |
| Error-maximizing rigid image transformations (Loosli et al., 2007; Fawzi et al., 2016) | $p(\theta)$ | Input | Adaptive stochastic & constrained, respectively | Training |
| Learning class-specific elastic image-deformation fields (Hauberg et al., 2016) | $p(\theta|t, \mathcal{D})$ | Input | Random | Training |
| Any handcrafted data preprocessing, for example scale-invariant feature transform (SIFT) for images (Lowe, 1999) | $p(\theta)$ | Input | Deterministic | Training and test |
| Overfeat (Sermanet et al., 2013) | $p(\theta)$ | Input | Deterministic | Training and test |

Table 2: Existing *domain-specific* data-based methods classified according to our taxonomy. Table columns are described in Section 3. Note that these methods are never applied on the hidden features, because domain knowledge cannot be applied on them.

bridge the limited-data gap between the expected and the empirical risk, Eqs. (2)–(3). While unlimited sampling from $Q$ provides more data than the original dataset $\mathcal{D}$, both of them usually are merely approximations of the ground truth data distribution or of an ideal training dataset; both $\mathcal{D}$ and $Q$ have their own distinct biases, advantages and disadvantages. For example, elastic image deformations result in images that are not perfectly realistic; this is not necessarily a disadvantage, but it is a bias compared to the ground truth data distribution; in any case, the advantages (having more training data) often prevail. In some cases, it may be even desired for $Q$ to be deliberately *different* from the ground truth data distribution. For example, in case of class imbalance (unbalanced abundance or importance of classes), a common regularization strategy is to undersample or oversample the data, sometimes leading to a less realistic $Q$ but better models. This is how an *ideal training dataset* may be different from the *ground truth data distribution*.

If the transformation is additionally **representation-preserving**, then the distribution $Q$ created by the transformation $\tau_\theta$ attempts to mimic the ground truth data distribution $P$. Otherwise, the notion of a "ground truth data distribution" in the modified representation may be vague. We provide more details about the transition from $\mathcal{D}$ to $Q$ in Appendix B.

**Summary of data-based methods**   Data-based regularization is a popular and very useful way to improve the results of deep learning. In this section we formalized this group of methods and showed that seemingly unrelated techniques such as Target-preserving data augmentation, Dropout, or Batch normalization are methodologically surprisingly close to each other. In Section 8 we discuss future directions that we find promising.

## 4   REGULARIZATION VIA THE NETWORK ARCHITECTURE

A network architecture $f$ can be selected to have certain properties or match certain assumptions in order to have a regularizing effect.[1]

---

[1]The network architecture is represented by a function $f : (w, x) \mapsto y$, and together with the set $W$ of all its possible weight configurations defines a set of mappings that this particular architecture can realize: $\{f_w : x \mapsto y \mid \forall w \in W\}$.

| Method | Method class | Assumptions about an appropriate learnable input-output mapping |
|---|---|---|
| Any chosen (not overly complex) architecture | * | Mapping can be well approximated by functions from the chosen family which are easily accessible by optimization. |
| Small network | * | Mapping is simple (complexity of the mapping depends on the number of network units and layers). |
| Deep network | * | The mapping is complex, but can be decomposed into a composition (or generally into a directed acyclic graph) of simple nonlinear transformations, e.g. affine transformation followed by simple nonlinearity (fully-connected layer), "multi-channel convolution" followed by simple nonlinearity (convolutional layer), etc. |
| Hard bottleneck (layer with few neurons); soft bottleneck (e.g. Jacobian penalty (Rifai et al., 2011c), see Section 6) | Layer operation | Data concentrates around a lower-dimensional manifold; has few factors of variation. |
| Convolutional networks (Fukushima and Miyake, 1982; Rumelhart et al., 1986, pp. 348-352; LeCun et al., 1989; Simard et al., 2003) | Layer operation | Spatially local and shift-equivariant feature extraction is all we need. |
| Dilated convolutions (Yu and Koltun, 2015) | Layer operation | Like convolutional networks. Additionally: Sparse sampling of wide local neighborhoods provides relevant information, and better preserves relevant high-resolution information than architectures with downscaling and upsampling. |
| Strided convolutions (see Dumoulin and Visin, 2016) | Layer operation | The mapping is reliable at reacting to features that do not vary too abruptly in space, i.e. which are present in several neighboring pixels and can be detected even if the filter center skips some of the pixels. The output is robust towards slight changes of the location of features, and changes of strength/presence of spatially strongly varying features. |
| Pooling | Layer operation | The output is invariant to slight spatial distortions of the input (slight changes of the location of (deep) features). Features that are sensitive to such distortions can be discarded. |
| Stochastic pooling (Zeiler and Fergus, 2013) | Layer operation | The output is robust towards slight changes of the location (like pooling) but also of the strength/presence of (deep) features. |
| Training with different kinds of noise (including Dropout; see Section 3) | Noise | The mapping is robust to noise: the given class of perturbations of the input or deep features should not affect the output too much. |
| Dropout (Hinton et al., 2012; Srivastava et al., 2014), DropConnect (Wan et al., 2013), and related methods | Noise | Extracting complementary (non-coadapted) features is helpful. Non-coadapted features are more informative, better disentangle factors of variation. (We want to disentangle factors of variation because they are entangled in different ways in inputs vs. in outputs.) When interpreted as ensemble learning: usual assumptions of ensemble learning (predictions of weak learners have complementary info and can be combined to strong prediction). |
| Maxout units (Goodfellow et al., 2013) | Layer operation | Assumptions similar to Dropout, with more accurate approximation of model averaging (when interpreted as ensemble learning) |
| Skip-connections (Long et al., 2015; Huang et al., 2016a) | Connections between layers | Certain lower-level features can directly be reused in a meaningful way at (several) higher levels of abstraction |
| Linearly augmented feed-forward network (van der Smagt and Hirzinger, 1998) | Connections between layers | Skip-connections that share weights with the non-skip-connections. Helps against vanishing gradients. Rather changes the learning algorithm than the network mapping. |
| Residual learning (He et al., 2016) | Connections between layers | Learning additive difference of a mapping $f$ (or its compositional parts) from the identity mapping is easier than learning $f$ itself. Meaningful deep features can be composed as a sum of lower-level and intermediate-level features. |
| Stochastic depth (Huang et al., 2016b), DropIn (Smith et al., 2015) | Connections between layers; noise | Similar to Dropout: extracting complementary (non-coadapted) features *across different levels of abstraction* is helpful; implicit model ensemble. Similar to Residual learning: meaningful deep features can be composed as a sum of lower-level and intermediate-level features, *with the intermediate-level ones being optional, and leaving them out being meaningful data augmentation.* Similar to Mollifying networks: simplifying random parts of the mapping improves training. |
| Mollifying networks (Gülçehre et al., 2016b) | Connections between layers; noise | The mapping can be easier approximated by estimating its decreasingly linear simplified version |
| Network information criterion (Murata et al., 1994), Network growing and network pruning (see Bishop, 1995a, Sec. 9.5) | Model selection | Optimal generalization is reached by a network that has the right number of units (not too few, not too many) |
| Multi-task learning (see Caruana, 1998; Ruder, 2017) | * | Several tasks can help each other to learn mutually useful feature extractors, as long as the tasks do not compete for resources (network capacity) |

Table 3: Methods based on network architecture, and rough description of assumptions that they encode. There are partial overlaps between some listed methods. For example, Residual learning uses Skip-connections. Many noise-based methods also fit Table 1 (cf. Appendix A).

**Assumptions about the mapping** An input-output mapping $f_w$ must have certain properties in order to fit the data $P$ well. Although it may be intractable to enforce the precise properties of an ideal mapping, it may be possible to approximate them by simplified assumptions about the mapping. These properties and assumptions can then be imposed upon model fitting in a hard or soft manner. This limits the search space of models and allows finding better solutions. An example is the decision about the number of layers and units, which allows the mapping to be neither too simple nor too complex (thus avoiding underfitting and overfitting). Another example are certain invariances of the mapping, such as locality and shift-equivariance of feature extraction hardwired in convolutional layers. Overall, the approach of imposing assumptions about the input-output mapping discussed in this section is the selection of the network architecture $f$. The choice of architecture $f$ on the one hand *hardwires* certain properties of the mapping; additionally, in an interplay between $f$ and the optimization algorithm (Section 7), certain weight configurations are more likely accessible by optimization than others, further limiting the likely search space in a *soft* way. A complementary way of imposing certain assumptions about the mapping are regularization terms (Section 6), as well as invariances present in the (augmented) data set (Section 3).

Assumptions can be hardwired into the definition of the *operation* performed by certain layers, and/or into the *connections* between layers. This distinction is made in Table 3, where these and other methods are listed.

In Section 3 about data, we mentioned regularization methods that transform data in the hidden-feature space. They can be considered part of the architecture. In other words, they fit both Sections 3 (data) and 4 (architecture). These methods are listed in Table 1 with *hidden features* as their transformation space.

**Weight sharing** Reusing a certain trainable parameter in several parts of the network is referred to as *weight sharing*. This usually makes the model less complex than using separately trainable parameters. An example are convolutional networks (LeCun et al., 1989). Here the weight sharing does not merely reduce the number of weights that need to be learned; it also encodes the prior knowledge about the shift-equivariance and locality of feature extraction. Another example is weight sharing in autoencoders.

**Activation functions** Choosing the right activation function is quite important; for example, using Rectified linear units (ReLUs) improved the performance of many deep architectures both in the sense of training times and accuracy as well as overcoming the need for greedy layer-wise pre-training (Hahnloser et al., 2000; Jarrett et al., 2009; Nair and Hinton, 2010; Glorot et al., 2011). The success of ReLUs can be partially attributed to the fact that they provide more expressive families of mappings compared to sigmoid activations (in the sense that the classical sigmoid nonlinearity can be approximated very well[2] with only two ReLUs, but it takes an infinite number of sigmoid units to approximate a ReLU) and their affine extrapolation to unknown regions of data space seems to provide better generalization in practice than the "stagnating" extrapolation of sigmoid units. However, their hard negative cut-off and unbounded positive part are not always desired properties. Some activation functions were designed explicitly for regularization. For Dropout, Maxout units (Goodfellow et al., 2013) allow a more precise approximation of the geometric mean of the model ensemble predictions at test time. Stochastic pooling (Zeiler and Fergus, 2013), on the other hand, is a noisy version of max-pooling. The authors claim that this allows modelling distributions of activations instead of taking just the maximum.

**Noisy models** Stochastic pooling was one example of a stochastic generalization of a deterministic model. Some models are stochastic by injecting random noise into various parts of the model. The most frequently used noisy model is Dropout (Hinton et al., 2012; Srivastava et al., 2014).

---

[2]Small integrated squared error, small integrated absolute error. A simple example is $\text{sigm}(x) \approx \text{ReLU}(x + 0.5) - \text{ReLU}(x - 0.5)$.

**Multi-task learning**  A special type of regularization is multi-task learning (see Caruana, 1998; Ruder, 2017), where the network is modified to predict targets for several tasks at once. It can be combined with semi-supervised learning to utilize unlabeled data on an auxiliary task (Rasmus et al., 2015). A similar concept of sharing knowledge between tasks is also utilized in *meta-learning*, where multiple tasks from the same domain are learned sequentially, using previously gained knowledge as bias for new tasks (Baxter, 2000); and *transfer learning*, where knowledge from one domain is transferred into another domain (Pan and Yang, 2010). These approaches differ from other methods in the sense that they require some additional target data, which are not always available.

**Model selection**  The best among several trained models (e.g. with different architectures) can be selected by evaluating the predictions on a validation set. It should be noted that this holds for selecting the best combination of all techniques (Sections 3–7), not just architecture; and that the validation set used for model selection in the "outer loop" should be different from the validation set used e.g. for Early stopping (Section 7), and different from the test set (Cawley and Talbot, 2010). However, there are also model selection methods that specifically target the selection of the *number of units* in a specific network architecture, e.g. using network growing and network pruning (see Bishop, 1995a, Sec. 9.5), or additionally do *not* require a validation set, e.g. the Network information criterion to compare models based on the training error and second derivatives of the loss function (Murata et al., 1994).

## 5  Regularization via the error function

Ideally, the error function $E$ reflects an appropriate notion of quality, and in some cases some assumptions about the data distribution. Typical examples are mean squared error or cross-entropy. The error function $E$ can also have a regularizing effect. An example is Dice coefficient optimization (Milletari et al., 2016) which is robust to class imbalance. Moreover, the overall form of the loss function can be different than Eq. (3). For example, in certain loss functions that are robust to class imbalance, the sum is taken over pairwise combinations $\mathcal{D} \times \mathcal{D}$ of training samples (Yan et al., 2003), rather than over training samples. But such alternatives to Eq. (3) are rather rare, and similar principles apply. If additional tasks are added for a regularizing effect (multi-task learning (see Caruana, 1998; Ruder, 2017)), then targets $t$ are modified to consist of several tasks, the mapping $f_w$ is modified to produce an according output $y$, and $E$ is modified to account for the modified $t$ and $y$. Besides, there are regularization terms that depend on $\partial E / \partial x$. They depend on $t$ and thus in our definition are considered part of $E$ rather than of $R$, but they are listed in Section 6 among $R$ (rather than here) for a better overview.

## 6  Regularization via the regularization term

Regularization can be achieved by adding a regularizer $R$ into the loss function. Unlike the error function $E$ (which expresses consistency of outputs with targets), the regularization term is independent of the targets. Instead, it is used to encode other properties of the desired model, to provide inductive bias (i.e. assumptions about the mapping other than consistency of outputs with targets). The value of $R$ can thus be computed for an unlabeled test sample, whereas the value of $E$ cannot.

The independence of $R$ from $t$ has an important implication: it allows additionally using unlabeled samples (semi-supervised learning) to improve the learned model based on its compliance with some desired properties (Sajjadi et al., 2016). For example, semi-supervised learning with ladder networks (Rasmus et al., 2015) combines a supervised task with an unsupervised auxiliary denoising task in a "multi-task" learning fashion. (For alternative interpretations, see Appendix A.) Unlabeled samples are extremely useful when labeled samples are scarce. A Bayesian perspective on the combination of labeled and unlabeled data in a semi-supervised manner is offered by Lasserre et al. (2006).

A classical regularizer is *weight decay* (see Plaut et al., 1986; Lang and Hinton, 1990; Goodfellow et al., 2016, Chap. 7):

$$R(w) = \lambda \frac{1}{2} \|w\|_2^2 \, , \tag{5}$$

where $\lambda$ is a weighting term controlling the importance of the regularization over the consistency. From the Bayesian perspective, weight decay corresponds to using a symmetric multivariate normal distribution as prior for the weights: $p(w) = \mathcal{N}(w|\mathbf{0}, \lambda^{-1}\mathbf{I})$ (Nowlan and Hinton, 1992). Indeed, $-\log \mathcal{N}(w|\mathbf{0}, \lambda^{-1}\mathbf{I}) \propto -\log \exp\left(-\frac{\lambda}{2}\|w\|_2^2\right) = \frac{\lambda}{2}\|w\|_2^2 = R(w)$. Weight decay has gained big popularity, and it is being successfully used; Krizhevsky et al. (2012) even observe reduction of the error on the *training* set.

Another common prior assumption that can be expressed via the regularization term is "smoothness" of the learned mapping (see Bengio et al., 2013, Section 3.2): if $x_1 \approx x_2$, then $f_w(x_1) \approx f_w(x_2)$. It can be expressed by the following loss term:

$$R(f_w, x) = \|J_{f_w}(x)\|_F^2 \, , \tag{6}$$

where $\|\cdot\|_F$ denotes the Frobenius norm, and $J_{f_w}(x)$ is the Jacobian of the neural network input-to-output mapping $f_w$ for some fixed network weights $w$. This term penalizes mappings with large derivatives, and is used in contractive autoencoders (Rifai et al., 2011c).

The domain of loss regularizers is very heterogeneous. We propose a natural way to categorize them *by their dependence*. We saw in Eq. (5) that weight decay depends on $w$ only, whereas the Jacobian penalty in Eq. (6) depends on $w$, $f$, and $x$. More precisely, the Jacobian penalty uses the derivative $\partial y/\partial x$ of output $y = f_w(x)$ w.r.t. input $x$. (We use vector-by-vector derivative notation from matrix calculus, i.e. $\partial y/\partial x = \partial f_w(x)/\partial x = J_{f_w}$ is the Jacobian of $f_w$ with fixed weights $w$.) We identify the following dependencies of $R$:

- Dependence on the weights $w$
- Dependence on the network output $y = f_w(x)$
- Dependence on the derivative $\partial y/\partial w$ of the output $y = f_w(x)$ w.r.t. the weights $w$
- Dependence on the derivative $\partial y/\partial x$ of the output $y = f_w(x)$ w.r.t. the input $x$
- Dependence on the derivative $\partial E/\partial x$ of the error term $E$ w.r.t. the input $x$ ($E$ depends on $t$, and according to our definition such methods belong to Section 5, but they are listed here for overview)

A review of existing methods can be found in Table 4. Weight decay seems to be still the most popular of the regularization terms. Some of the methods are equivalent or nearly equivalent to other methods from different taxonomy branches. For example, Tangent prop simulates minimal data augmentation (Simard et al., 1992); Injection of small-variance Gaussian noise (Bishop, 1995b; An, 1996) is an approximation of Jacobian penalty (Rifai et al., 2011c); and Fast dropout (Wang and Manning, 2013) is (in shallow networks) a deterministic approximation of Dropout. This is indicated in the *Equivalence* column in Table 4.

## 7 Regularization via optimization

The last class of the regularization methods according to our taxonomy is the regularization through optimization. While this may sound unusual, optimization and regularization cannot be clearly separated in the context of deep learning where it is not so crucial what the optimum of the empirical risk is (because it cannot be found exactly, and the ultimate goal is minimizing the *expected* risk anyway). Instead, the shape of the loss function and the optimization procedure play together to dictate how the training proceeds in the weight space and where it ends up. To demonstrate the overlap of regularization and optimization, we show in Figure 1 how one of the most prominent regularization methods, Dropout, can be seen as a modification of the optimization procedure.

Stochastic gradient descent (SGD) (see Bottou, 1998) (along with its derivations) is the most frequently used optimization algorithm in the context of deep neural networks and is the center of our attention. We also list some alternative methods below.

| Method | Description | Dependency | | | | | Equivalence |
|---|---|---|---|---|---|---|---|
| | | $w$ | $y$ | $\frac{\partial y}{\partial w}$ | $\frac{\partial y}{\partial x}$ | $\frac{\partial E}{\partial x}$ | |
| Weight decay (see Plaut et al., 1986; Lang and Hinton, 1990; Goodfellow et al., 2016, Chap. 7) | $L^2$ norm on network weights (not biases). Favors smaller weights, thus for usual architectures tends to make the mapping less "extreme", more robust to noise in the input. | ✖ | | | | | Early stopping (see Collobert and Bengio, 2004; Goodfellow et al., 2016, Chap. 7) |
| Weight smoothing (Lang and Hinton, 1990) | Penalizes $L^2$ norm of gradients of learned filters, making them smooth. Not beneficial in practice. | ✖ | | | | | |
| Weight elimination (Weigend et al., 1991) | Similar to weight decay but favors few stronger connections over many weak ones. | ✖ | | | | | Goal similar to Narrow and broad Gaussians |
| Soft weight-sharing (Nowlan and Hinton, 1992) | Mixture-of-Gaussians prior on weights. Generalization of weight decay. Weights are pushed to form a predefined number of groups with similar values. | ✖ | | | | | |
| Narrow and broad Gaussians (Nowlan and Hinton, 1992; Blundell et al., 2015) | Weights come from two Gaussians, a narrow and a broad one. Special case of Soft weight-sharing. | ✖ | | | | | Goal similar to Weight elimination |
| Fast dropout approximation (Wang and Manning, 2013) | Approximates the loss that dropout minimizes. Weighted $L^2$ weight penalty. Only for shallow networks. | ✖ | ✖ | | | | Dropout |
| Mutual exclusivity (Sajjadi et al., 2016) | Unlabeled samples push decision boundaries to low-density regions in input space, promoting sharp (confident) predictions. | | ✖ | | | | |
| Segmentation with binary potentials (BenTaieb and Hamarneh, 2016) | Penalty on anatomically implausible image segmentations. | | ✖ | | | | |
| Flat minima search (Hochreiter and Schmidhuber, 1995) | Penalty for sharp minima, i.e. for weight configurations where small weight perturbation leads to high error increase. Flat minima have low Minimum description length (i.e. exhibit ideal balance between training error and model complexity) and thus should generalize better (Rissanen, 1986). | ✖ | | ✖ | | | |
| Tangent prop (Simard et al., 1992) | $L^2$ penalty on directional derivative of mapping in the predefined *tangent directions* that correspond to known input-space transformations. | | | | ✖ | | Simple data augmentation |
| Jacobian penalty (Rifai et al., 2011c) | $L^2$ penalty on the Jacobian of (parts of) the network mapping—smoothness prior. | | | | ✖ | | Noise on inputs injection (not exact (see An, 1996)) |
| Manifold tangent classifier (Rifai et al., 2011a) | Like tangent prop, but the input "tangent" directions are extracted from manifold learned by a stack of contractive autoencoders and then performing SVD of the Jacobian at each input sample. | | | | ✖ | | |
| Hessian penalty (Rifai et al., 2011b) | Fast way to approximate $L^2$ penalty of the Hessian of $f$ by penalizing Jacobian with noisy input. | | | | ✖ | | |
| Tikhonov regularizers (Bishop, 1995b) | $L^2$ penalty on (up to) $n$-th derivative of the learned mapping w.r.t. input. | | | | ✖ | | For penalty on first derivative: noise on inputs injection (not exact (see An, 1996)) |
| Loss-invariant backpropagation (Demyanov et al., 2015, Sec. 3.1; Lyu et al., 2015) | $(L^2)$ norm of gradient of loss w.r.t. input. Changes the mapping such that the loss becomes rather invariant to changes of the input. | | | | | ✖ | Adversarial training |
| Prediction-invariant backpropagation (Demyanov et al., 2015, Sec. 3.2) | $(L^2)$ norm of directional derivative of mapping w.r.t. input in the direction of $x$ causing the largest increase in loss. | | | | ✖ | ✖ | Adversarial training |

Table 4: Regularization terms, with dependencies marked by ✖. Methods that depend on $\partial E/\partial x$ implicitly depend on targets $t$ and thus can be considered part of the error function (Section 5) rather than regularization term (Section 6).

Stochastic gradient descent is an iterative optimization algorithm using the following update rule:

$$w_{t+1} = w_t - \eta_t \nabla_w \mathcal{L}(w_t, d_t),$$ (7)

where $\nabla \mathcal{L}(w_t, d_t)$ is the gradient of the loss $\mathcal{L}$ evaluated on a mini-batch $d_t$ from the training set $\mathcal{D}$. It is frequently used in combination with *momentum* and other tweaks improving the convergence speed (see Wilson et al., 2017). Moreover, the noise induced by the varying mini-batches helps the algorithm escape saddle points (Ge et al., 2015); this can be further reinforced by adding supplementary gradient noise (Neelakantan et al., 2015; Chaudhari and Soatto, 2015).

If the algorithm reaches a low training error in a reasonable time (linear in the size of the training set, allowing multiple passes through $\mathcal{D}$), the solution generalizes well under certain mild assumptions; in that sense SGD works as an *implicit regularizer*: a short training time prevents overfitting even without any additional regularizer used (Hardt et al., 2016). This is in line with (Zhang et al., 2017) who find in a series of experiments that regularization (such as Dropout, data augmentation, and weight decay) is by itself neither necessary nor sufficient for good generalization.

We divide the methods into three groups: initialization/warm-start methods, update methods, and termination methods, discussed in the following.

**Initialization and warm-start methods**   These methods affect the initial selection of the model weights. Currently the most frequently used method is sampling the initial weights from a carefully tuned distribution. There are multiple strategies based on the architecture choice, aiming at keeping the variance of activations in all layers around 1, thus preventing vanishing or exploding activations (and gradients) in deeper layers (Glorot and Bengio, 2010, Sec. 4.2; He et al., 2015).

Another (complementary) option is *pre-training* on different data, or with a different objective, or with partially different architecture. This can prime the learning algorithm towards a good solution before the fine-tuning on the actual objective starts. Pre-training the model on a different task in the same domain may lead to learning useful features, making the primary task easier. However, pre-trained models are also often misused as a lazy approach to problems where training from scratch or using thorough domain adaptation, transfer learning, or multi-task learning methods would be worth trying. On the other hand, pre-training or similar techniques may be a useful *part* of such methods.

Finally, with some methods such as Curriculum learning (Bengio et al., 2009), the transition between pre-training and fine-tuning is smooth. We refer to them as *warm-start methods*.

- Initialization without pre-training
    - Random weight initialization (Rumelhart et al., 1986, p. 330; Glorot and Bengio, 2010; He et al., 2015; Hendrycks and Gimpel, 2016)
    - Orthogonal weight matrices (Saxe et al., 2013)
    - Data-dependent weight initialization (Krähenbühl et al., 2015)
- Initialization with pre-training
    - Greedy layer-wise pre-training (Hinton et al., 2006; Bengio et al., 2007; Erhan et al., 2010) (has become less important due to advances (e.g. ReLUs) in effective end-to-end training that optimizes all parameters simultaneously)
    - Curriculum learning (Bengio et al., 2009)
    - Spatial contrasting (Hoffer et al., 2016)
    - Subtask splitting (Gülçehre and Bengio, 2016)

**Update methods**   This class of methods affects individual weight updates. There are two complementary subgroups: *Update rules* modify the form of the update formula; *Weight and gradient filters* are methods that affect the value of the gradient or weights, which are used in the update formula, e.g. by injecting noise into the gradient (Neelakantan et al., 2015).

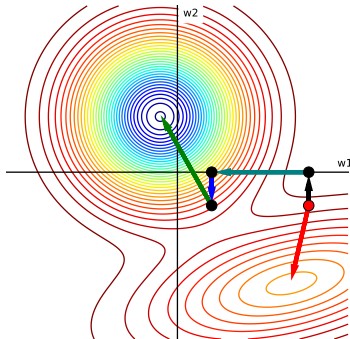

Figure 1: Effect of Dropout on weight optimization. Starting from the current weight configuration (red dot), all weights of certain neurons are set to zero (black arrow), descent step is performed in that subspace (teal arrow), and then the discarded weight-space coordinates are restored (blue arrow).

Again, it is not entirely clear which of the methods only speed up the optimization and which actually help the generalization. Wilson et al. (2017) show that some of the methods such as AdaGrad or Adam even lose the regularization abilities of SGD.

- Update rules
    - Momentum, Nesterov's accelerated gradient method, AdaGrad, AdaDelta, RMSProp, Adam—overview in (Wilson et al., 2017)
    - Learning rate schedules (Girosi et al., 1995; Hoffer et al., 2017)
    - Online batch selection (Loshchilov and Hutter, 2015)
    - SGD alternatives: L-BFGS (Liu and Nocedal, 1989; Le et al., 2011), Hessian-free methods (Martens, 2010), Sum-of-functions optimizer (Sohl-Dickstein et al., 2014), ProxProp (Frerix et al., 2017)
- Gradient and weight filters
    - Annealed Langevin noise (Neelakantan et al., 2015)
    - AnnealSGD (Chaudhari and Soatto, 2015)
    - Dropout (Hinton et al., 2012; Srivastava et al., 2014) corresponds to optimization steps in subspaces of weight space, see Figure 1
    - Annealed noise on targets (Wang and Principe, 1999) (works as noise on gradient, but belongs rather to data-based methods, Section 3)

**Termination methods**   There are numerous possible stopping criteria and selecting the right moment to stop the optimization procedure may improve the generalization by reducing the error caused by the discrepancy between the minimizers of expected and empirical risk: The network first learns general concepts that work for all samples from the ground truth distribution $P$ before fitting the specific sample $\mathcal{D}$ and its noise (Krueger et al., 2017).

The most successful and popular termination methods put a portion of the labeled data aside as a *validation set* and use it to evaluate performance (*validation error*). The most prominent example is Early stopping (see Prechelt, 1998). Collobert and Bengio (2004) show that Early stopping has the same effect as Weight decay regularization penalty term in multi-layered perceptrons with linear output units; however, its hyperparameters are easier to tune.

In scenarios where the training data are scarce it is possible to resort to termination methods that do *not* use a validation set. The simplest case is fixing the number of passes through the training set.

- Termination using a validation set
    - Early stopping (see Morgan and Bourlard, 1990; Prechelt, 1998)
    - Choice of validation set size based on test set size (Amari et al., 1997)
- Termination *without* using a validation set
    - Fixed number of iterations
    - Optimized approximation algorithm (Liu et al., 2008)

## 8 Recommendations, discussion, conclusions

We see the main benefits of our taxonomy to be two-fold: Firstly, it provides an overview of the existing techniques to the users of regularization methods and gives them a better idea of how to choose the ideal combination of regularization techniques for their problem. Secondly, it is useful for development of new methods, as it gives a comprehensive overview of the main principles that can be exploited to regularize the models. We summarize our recommendations[3] in the following paragraphs:

**Recommendations for users of existing regularization methods** Overall, using the information contained in data as well as prior knowledge as much as possible, and primarily starting with popular methods, the following procedure can be helpful:

- Common recommendations for the first steps:
  - Deep learning is about disentangling the factors of variation. An appropriate data representation should be chosen; *known* meaningful data transformations should *not* be outsourced to the learning. Redundantly providing the same information in several representations is okay.
  - Output nonlinearity and error function should reflect the learning goals.
  - A good starting point are techniques that usually work well (e.g. ReLU, successful architectures). Hyperparameters (and architecture) can be tuned jointly, but "lazily" (interpolating/extrapolating from experience instead of trying too many combinations).
  - Often it is helpful to start with a simplified dataset (e.g. fewer and/or easier samples) and a simple network, and after obtaining promising results gradually increasing the complexity of both data and network while tuning hyperparameters and trying regularization methods.
- Regularization via data:
  - When not working with nearly infinite/abundant data:
    * Gathering more real data (and using methods that take its properties into account) is advisable if possible:
      · Labeled samples are best, but unlabeled ones can also be helpful (compatible with semi-supervised learning).
      · Samples from the same domain are best, but samples from similar domains can also be helpful (compatible with domain adaptation and transfer learning).
      · Reliable high-quality samples are best, but lower-quality ones can also be helpful (their confidence/importance can be adjusted accordingly).
      · Labels for an additional task can be helpful (compatible with multi-task learning).
      · Additional input features (from additional information sources) and/or data preprocessing (i.e. domain-specific data transformations) can be helpful (the network architecture needs to be adjusted accordingly).
    * Data augmentation (e.g. target-preserving handcrafted domain-specific transformations) can well compensate for limited data. If natural ways to augment data (to mimic natural transformations sufficiently well) are known, they can be tried (and combined).
    * If natural ways to augment data are unknown or turn out to be insufficient, it may be possible to infer the transformation from data (e.g. learning image-deformation fields) if a sufficient amount of data is available for that.
  - Popular generic methods (e.g. advanced variants of Dropout) often also help.
- Architecture and regularization terms:

---

[3]Note that these recommendations are neither the only nor the best way; every dataset may require a slightly different approach. Our recommendations are a summary of what we found to work well, and what seems to be common themes and "written between the lines" in many state-of-the-art works.

- Knowledge about possible meaningful properties of the mapping can be used to e.g. hardwire invariances (to certain transformations) into the architecture, or be formulated as regularization terms.
- Popular methods may help as well (see Tables 3–4), but should be chosen to match the assumptions about the mapping (e.g. convolutional layers are fully appropriate only if local and shift-equivariant feature extraction on regular-grid data is desired).

- Optimization:
    - Initialization: Even though pre-trained ready-made models greatly speed up prototyping, training from a good random initialization should also be considered.
    - Optimizers: Trying a few different ones, including advanced ones (e.g. Nesterov momentum, Adam, ProxProp), may lead to improved results. Correctly chosen parameters, such as learning rate, usually make a big difference.

**Recommendations for developers of novel regularization methods**   Getting an overview and understanding the reasons for the success of the best methods is a great foundation. Promising empty niches (certain combinations of taxonomy properties) exist that can be addressed. The assumptions to be imposed upon the model can have a strong impact on most elements of the taxonomy. Data augmentation is more expressive than loss terms (loss terms enforce properties only in infinitesimally small neighborhood of the training samples; data augmentation can use rich transformation parameter distributions). Data and loss terms impose assumptions and invariances in a rather soft manner, and their influence can be tuned, whereas hardwiring the network architecture is a harsher way to impose assumptions. Different assumptions and options to impose them have different advantages and disadvantages.

**Future directions for data-based methods**   There are several promising directions that in our opinion require more investigation: Adaptive sampling of $\theta$ might lead to lower errors and shorter training times (Fawzi et al., 2016) (in turn, shorter training times may additionally work as implicit regularization (Hardt et al., 2016), see also Section 7). Secondly, learning class-dependent transformations (i.e. $p(\theta|t)$) in our opinion might lead to more plausible samples. Furthermore, the field of adversarial examples (and network robustness to them) is gaining increased attention after the recently sparked discussion on real-world adversarial examples and their robustness/invariance to transformations such as the change of camera position (Lu et al., 2017; Athalye and Sutskever, 2017). Countering strong adversarial examples may require better regularization techniques.

**Summary**   In this work we proposed a broad definition of regularization for deep learning, identified five main elements of neural network training (data, architecture, error term, regularization term, optimization procedure), described regularization via each of them, including a further, finer taxonomy for each, and presented example methods from these subcategories. Instead of attempting to explain referenced works in detail, we merely pinpointed their properties relevant to our categorization. Our work demonstrates some links between existing methods. Moreover, our systematic approach enables the discovery of new, improved regularization methods by combining the best properties of the existing ones.

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

## A   Ambiguities in the taxonomy

Although our proposed taxonomy seems intuitive, there are some ambiguities: Certain methods have multiple interpretations matching various categories. Viewed from the exterior, a neural network maps inputs $x$ to outputs $y$. We formulate this as $y = f_w(\tau_\theta(x))$ for transformations $\tau_\theta$ in input space (and similarly for hidden-feature space, where $\tau_\theta$ is applied in between layers of the network $f_w$). However, how to split this $x$-to-$y$ mapping into "the $\tau_\theta$ part" and "the $f_w$ part", and thus into Section 3 vs. Section 4, is ambiguous and up to one's taste and goals. In our choices (marked with "☑" below), we attempt to use common notions and Occam's razor.

- Ambiguity of attributing noise to $f$, or to $w$, or to data transformations $\tau_\theta$:
  - Stochastic methods such as Stochastic depth (Huang et al., 2016b) can have several interpretations if stochastic transformations are allowed for $f$ or $w$:
    - ☑ Stochastic transformation of the architecture $f$ (randomly dropping some connections), Table 3
    - ☐ Stochastic transformation of the weights $w$ (setting some weights to 0 in a certain random pattern)
    - ☐ Stochastic transformation $\tau_\theta$ of data in hidden-feature space; dependence is $p(\theta)$, described in Table 1 for completeness
- Ambiguity of splitting $\tau_\theta$ into $\tau$ and $\theta$:
  - Dropout:
    - ☑ Parameters $\theta$ are the dropout mask; dependence is $p(\theta)$; transformation $\tau$ applies the dropout mask to the hidden features

☐ Parameters $\theta$ are the seed state of a pseudorandom number generator; dependence is $p(\theta)$; transformation $\tau$ internally generates the random dropout mask from the random seed and applies it to the hidden features

– Projecting dropout noise into input space (Bouthillier et al., 2015, Sec. 3) can fit our taxonomy in different ways by defining $\tau$ and $\theta$ accordingly. It can have similar interpretations as Dropout above (if $\tau$ is generalized to allow for dependence on $x, f, w$), but we prefer the third interpretation without such generalizations:

☐ Parameters $\theta$ are the dropout mask (to be applied in a hidden layer); dependence is $p(\theta)$; transformation $\tau$ transforms the input to mimic the effect of the mask

☐ Parameters $\theta$ are the seed state of a pseudorandom number generator; dependence is $p(\theta)$; transformation $\tau$ internally generates the random dropout mask from the random seed and transforms the input to mimic the effect of the mask

☑ Parameters $\theta$ describe the transformation of the input in any formulation; dependence is $p(\theta|x, f, w)$; transformation $\tau$ merely applies the transformation in input space

- Ambiguity of splitting the network operation $f_w$ into layers: There are several possibilities to represent a function (neural network) as a composition (or directed acyclic graph) of functions (layers).

- Many of the input and hidden-feature transformations (Section 3) can be considered layers of the network (Section 4). In fact, the term "layer" is not uncommon for Dropout or Batch normalization.

- The usage of a trainable parameter in *several* parts of the network is called weight sharing. However, some mappings can be expressed with two equivalent formulas such that a parameter appears only once in one formulation, and several times in the other.

- Ambiguity of $E$ vs. $R$: Auxiliary denoising task in ladder networks (Rasmus et al., 2015) and similar autoencoder-style loss terms can be interpreted in different ways:

☑ Regularization term $R$ without given auxiliary targets $t$

☐ The ideal reconstructions can be considered as targets $t$ (if the definition of "targets" is slightly modified) and thus the denoising task becomes part of the error term $E$

## B    DATA-AUGMENTED LOSS FUNCTION

To understand the success of target-preserving data augmentation methods, we consider the *data-augmented loss function*, which we obtain by replacing the training samples $(x_i, t_i) \in \mathcal{D}$ in the empirical risk loss function (Eq. (3)) by augmented training samples $(\tau_\theta(x_i), t_i)$:

$$
\begin{aligned}
\hat{\mathcal{L}}_A &= \frac{1}{|\mathcal{D}|} \sum_{(x_i, t_i) \in \mathcal{D}} \mathbb{E}_\theta \Big[ \ell\big(\tau_\theta(x_i), t_i\big) \Big] \\
&= \frac{1}{|\mathcal{D}|} \sum_{(x_i, t_i) \in \mathcal{D}} \int_\Theta \Big( \ell\big(\tau_\theta(x_i), t_i\big) \Big) p(\theta) \, \mathrm{d}\theta,
\end{aligned}
\tag{8}
$$

where we have replaced the inner part ($E$ and $R$) of the loss function by $\ell$ to simplify the notation. Moreover, $\hat{\mathcal{L}}_A$ can be rewritten as

$$
\begin{aligned}
\hat{\mathcal{L}}_A &= \iint\limits_{X,T} \frac{1}{|\mathcal{D}|} \sum_{(x_i,t_i)\in\mathcal{D}} \int_{\Theta} \ell(x,t)\, p(\theta)\, \delta\big(x - \tau_\theta(x_i)\big)\, \delta(t - t_i)\, \mathrm{d}\theta\, \mathrm{d}t\, \mathrm{d}x \\
&= \iint\limits_{X,T} \ell(x,t) \left[ \frac{1}{|\mathcal{D}|} \sum_{(x_i,t_i)\in\mathcal{D}} \int_{\Theta} \delta\big(x - \tau_\theta(x_i)\big)\, \delta(t - t_i)\, p(\theta)\, \mathrm{d}\theta \right] \mathrm{d}t\, \mathrm{d}x \\
&= \iint\limits_{X,T} \ell(x,t)\, q(x,t)\, \mathrm{d}t\, \mathrm{d}x,
\end{aligned}
\tag{9}
$$

where $\delta(x)$ is the Dirac delta function: $\delta(x) = 0 \,\forall\, x \neq 0$ and $\int \delta(x)\,\mathrm{d}x = 1$; and $q(x,t)$ is defined as

$$
q(x,t) = \frac{1}{|\mathcal{D}|} \sum_{(x_i,t_i)\in\mathcal{D}} \int_{\Theta} \delta\big(x - \tau_\theta(x_i)\big)\, \delta(t - t_i)\, p(\theta)\, \mathrm{d}\theta.
\tag{10}
$$

Since $q$ is non-negative and $\iint q(x,t)\,\mathrm{d}x\,\mathrm{d}t = 1$, it is a valid probability density function inducing the distribution $Q$ of augmented data. Therefore,

$$
\hat{\mathcal{L}}_A = \mathbb{E}_{(x,t)\sim Q}\big[\ell(x,t)\big].
\tag{11}
$$

When $Q = P$, Eq. (11) becomes the expected risk (2). We can show how this is related to *importance sampling*:

$$
\begin{aligned}
\mathcal{L} &= \mathbb{E}_{(x,t)\sim P}\big[\ell(x,t)\big] \\
&= \iint\limits_{X,T} \ell(x,t)p(x,t)\,\mathrm{d}t\,\mathrm{d}x \\
&= \iint\limits_{X,T} \ell(x,t)\frac{p(x,t)}{q(x,t)}q(x,t)\,\mathrm{d}t\,\mathrm{d}x \\
&= \mathbb{E}_{(x,t)\sim Q}\left[\ell(x,t)\frac{p(x,t)}{q(x,t)}\right] \\
&\neq \mathbb{E}_{(x,t)\sim Q}\big[\ell(x,t)\big] \\
&= \hat{\mathcal{L}}_A.
\end{aligned}
\tag{12}
$$

The difference between $\mathcal{L}$ and $\hat{\mathcal{L}}_A$ is the re-weighting term $p(x,t)/q(x,t)$ identical to the one known from importance sampling (see Bishop, 1995a). The more similar $Q$ is to $P$ (i.e. the closer $Q$ models the ground truth distribution $P$), the more similar the augmented-data loss $\hat{\mathcal{L}}_A$ is to the expected loss $\mathcal{L}$. We see that data augmentation tries to simulate the real distribution $P$ by creating new samples from the training set $\mathcal{D}$, bridging the gap between the expected and the empirical risk.

