# OpenReview forum: "Regularization for Deep Learning: A Taxonomy"
_ICLR.cc/2018/Conference — Reject_

### Official Review · AnonReviewer1 · 2017-11-24
**Review of 'Regularization for DL: a taxonomy'**

**Rating:** 5
**Confidence:** 5

**Review:**

This paper is unusual in that it is more of a review than contributing novel knowledge. It considers a taxonomy of all the ways that machine learning (mostly deep learning) methods can achieve a form of regularization.

Unfortunately, it starts with a definition of regularization ('making the model generalize better') which I believe misses the point which was made in Goodfellow et al 2016 ('intend to improve test error but not necessarily training error'), i.e., that we would like to separate as much as possible the regularization effects from the optimization effect. Indeed, under the definition proposed here, any improvement in the optimizer could be considered like a regularizer, so long as we are not in the overfitting regime. That does not sound right to me.

There are several places where the authors make TOO STRONG STATEMENTS, taking for truth what are simply beliefs with no strong supporting evidence (at least published). This is not good for a review and when making recommendations.

The other weakness I estimate in this paper is that I did not get a sense that the taxonomy really helped us (me at least) to get insight into the different mentions being cited. Besides the obvious proposal to combine ideas to write new papers (but we did not need that paper to figure that out) I did not find much meat in the 'future directions' section.

However, I that except in a few places the understand of the field displayed by the authors is pretty good and, with correction, could serve as a useful reference for students of deep learning. The recommendations were reasonable although lacking empirical support (or pointers to the literature), so I would take them somewhat carefully, more as the current 'group think' than ground truth.

Finally, here a few minor points which could be fixed.

Eq. 1: in typical DL, minimization is approximate, not exact, so the proposed formalism does not reflect reality.

Eq. 4: in many cases, the noise is not added (e.g. dropout), so that should be clarified there.

page 3, first bullet of 'Effect on the data representation': not clear, may want to give translations as an example of such transformations,.

page 8, activation functions: the ReLU is actually older than the cited papers, it was used by computational neuroscientists a long time ago. Jarrett 2009 did not use the ReLU but an absolute-value rectifier and it was Glorot 2011 who showed that the ReLU really kicked ass for deeper networks. Nair 2010 used the ReLU in a very different context (RBMs), not really feedforward multi-layer networks where it shines now.
In that same section (and probably elsewhere) there are TOO STRONG STATEMENTS, e.g., the "facts" mentioned are not facts but merely folk belief, as far as I know, and I would like to see well-done supporting evidence before treating those as facts. Note that approximating the sigmoid precisely would require many ReLUs!

page 8: it is not clear how multi-task learning fits under the 'architecture' formalism provided at the beginning of section 4.

section 7 (page 10): there is earlier work on the connection between early stopping and L2 regularization, at least dating back to Ronan Collobert's PhD thesis (with neural nets), probably earlier for linear systems.

---

> ### Author Response · Authors · 2018-01-04
> **author rebuttal - additional minor comments to R1**
>
> Major comments are in our common response to all reviewers.
> Minor comments to R1:
> 1. Regarding too strong statements:
> We tried our best to provide references for all the statements made in our paper, allowing the readers to find out the precise context from which each claim originates. In the revised version, we removed the problematic statement about the effect of ReLUs on vanishing gradients and clarified the statement about their expressivity. If there are still some other too strong statements, it would be most helpful if the reviewer could identify them and we will be glad to adjust them.
>
> 2. "I did not get a sense that the taxonomy really helped us (me at least) to get insight into the different mentions being cited. "
> As we mention both in the abstract and in the introduction, we did not attempt to fully describe all details of the individual listed methods: "We are aware that the research works discussed in this taxonomy cannot be summarized in a single sentence. For the sake of structuring the multitude of papers, we decided to merely describe a certain subset of their properties according to the focus of our taxonomy."
>
> Instead, we aimed to identify the tools" they rely on and to show the connections between them. We provide insights about atomic properties of methods (e.g. description of possible data transformations on pages 5-6), and about different ways to interpret certain methods (e.g. Figure 1).
>
> 3. "The recommendations were reasonable although lacking empirical support (or pointers to the literature), so I would take them somewhat carefully, more as the current 'group think' than ground truth."
> Describing methods without saying when and how to use them is inconclusive and may leave many readers more confused than informed, which is why we added this section. Moreover, providing some practical pointers and recommended approaches improves the application potential of our paper and increases its value for readers with limited experience with deep learning.
>
> We agree that these recommendations are primarily based on our experience and general "unwritten knowledge" that is "between the lines" in state-of-the-art literature; to make this clear, we added this following disclaimer into the text: "Note that these recommendations are neither the only nor the best way; every dataset may require a slightly different approach. Our recommendations are a summary of what we found to work well, and what seems to be common themes and "written between the lines" in many state-of-the-art works"
>
> 4. Eq. 1 was updated.
>
> 5. Eq. 4: As mentioned in its preceding sentence, this equation gives merely an example of a transformation. Indeed, it can have any other, more complicated form.
>
> 6. Effect on data representation:
> We intended to keep this list free of examples and believe the explanations are clear enough. Examples like translation transformation might also mislead the reader about what exact property we mean and introduce false idea of necessary rigidity.
>
> 7. Regarding activation functions:
> In the revised version, we added a reference to (Hahnloser et al., 2000). The other papers are cited for the following reasons: Jarett et al. (2009) is the first occurrence in deep learning context ("Several rectifying non-linearities were tried, *including the positive part*, and produced similar results."), Nair and Hinton (2010) are the first to call the function "ReLU", and finally Glorot et al. (2011) is mentioned exactly for the reasons stated by the reviewer, as a very good overview of the properties and qualities of this activation.
>
> 8. "Note that approximating the sigmoid precisely would require many ReLUs!"
> We do not claim it can be done precisely, instead we give an example of an approximation with small integrated absolute error and small integrated squared error. Such small/finite integrated absolute error and integrated squared error are possible when approximating a sigmoid with few ReLUs, but not when approximating a ReLU with few sigmoids. Note that similar approximation of tanh (hard tanh) is often used in practice.
>
> 9. Regarding multi-task learning:
> We included multi-task learning in the architecture section because the network architecture needs to be modified (additional branches etc.) to process additional tasks. Note that we also mention it in the error function section because also the error function needs to be modified. We updated the text to make this clear.
>
> 10. Regarding the discussion on page 10:
> The discussion on page 10 is not related to the work of Collobert and Bengio (2004), who analyze the connection between early stopping and L2 regularization. Our discussion focuses on properties of SGD and the relation between training and testing error whereas early stopping relies on the connection between validation and test error. However, we appreciate the remark about the connection between early stopping and L2 regularization, and we added it to the article.
>
> See also response to all reviewers.

---

### Official Review · AnonReviewer2 · 2017-11-28

**Rating:** 4
**Confidence:** 4

**Review:**

The paper attempts to build a taxonomy for regularization techniques employed in deep learning. The authors categorize existing works related to regularization into five big categories, including data, model architecture, regularization term, error term and optimization. Subgroups are identified based on certain attributes.

The paper is written as a survey paper on literatures related to regularization in deep learning. The five top-level categories are quite obvious.  The authors organize works belonging to the first three categories into three big tables, and summarizing the key point of each one using one-liners to provide an overview for readers. While it is a worthy effort, I am not sure it offers much value to readers. Also there is a mix of trainability and regularization. Some of the works were proposed to address trainability issues instead of regularization, for example, densenet, and some of the initialization techniques.

The authors try to group items in each category into sub-groups according to certain attributes, however,  little explanation on how and why these attributes are identified was provided. For example, in table 1, what kind of information does the transformation space or phase provide in terms of helping readers choosing a particular data transformation / augmentation technique. At the end of section 3, the authors claim that dropout, BN are close to each other. Please elaborate on this point.

The authors offer some recommendation on how to choose or combine different regularization techniques at the end of the paper. However, it is not clear from reading the paper where these insights came from.

---

> ### Author Response · Authors · 2018-01-04
> **author rebuttal - additional minor comments to R2**
>
> Major comments can be found in our common response to all reviewers.
> Minor comments to R2:
> 1. "The authors try to group items in each category into sub-groups according to certain attributes, however, little explanation on how and why these attributes are identified was provided..."
> In sections 6 and 7 we provided clear explanations about the choice of subcategories. The remaining sections do not allow such clear distinction and our subcategories are one of several possible choices. Our choice is driven by the goal of separating as many separable concepts as possible.
>
> 2. "For example, in table 1, what kind of information does the transformation space or phase provide in terms of helping readers choosing a particular data transformation / augmentation technique"
> The taxonomy is not only about heuristics for choosing among methods. It is about something more fundamental: about understanding the "atomic" properties of the methods and their relationships. The transformation space and the phase are properties of methods. We do not claim that understanding the properties fully dictates which method will work well with what dataset.
>
> 3. Regarding the closeness between Dropout and Batch normalization:
> Here we refer to the fact that both methods rely on applying a simple transformation on the hidden-feature representation of the data.
>
> 4. "The authors offer some recommendation on how to choose or combine different regularization techniques at the end of the paper. However, it is not clear from reading the paper where these insights came from."
> See #3 in the comments to R1
>
> Major comments can be found in our common response to all reviewers.

---

### Official Review · AnonReviewer3 · 2017-12-04
**interetsing review but not appropriate for ICLR proceedings**

**Rating:** 4
**Confidence:** 5

**Review:**

The main aim of ICLR conference, at least as it is written on its website, is to provide new results on theories, methods and algorithms, supporting further breakthroughs in AI and DL.

In this respect the authors of the paper claim that their “systematic approach enables the discovery of new, improved regularization methods by combining the best properties of the existing ones.”

However, the authors did not provide any discoveries concerning new approaches to regularisation supporting this claim. Thus, the main contribution of the paper is that the authors made a review and performed classification of available regularisation methods. So, the paper is in fact a survey paper, which is more appropriate for full-scale journals. The work, developed by the authors, is really big. However, I am not sure it will bring a lot of benefits for readers except those who need review for some reports, introductions in PhD thesis, etc.

Although the authors mentioned some approaches to combine different regularisations, they did not performed any experiments supporting their ideas.

Thus, I think that
- the paper is well written in general,
- it can be improved (by taking into account several important comments from the Reviewer 2) and served as a review paper in some appropriate journal,
- the paper is not suited for ICLR proceedings due to reasons, mentioned above.

---

> ### Author Response · Authors · 2018-01-04
> **author rebuttal - additional minor comments to R3**
>
> Major comments can be found in our common response to all reviewers.
> Minor comments to R3:
> 1. "Although the authors mentioned some approaches to combine different regularisations, they did not perform any experiments supporting their ideas."
> The core of our work was designing the taxonomy and identification of the atomic building blocks of individual regularization methods. Designing new types of regularization and validating them experimentally was not our aim, which is why these hints are in the section "Future directions". There is a vast amount of possibilities to recombine in novel ways the atomic properties which we described; this would go beyond the scope of our work.
>
> Major comments can be found in our common response to all reviewers.

---

### Author Response · Authors · 2018-01-04
**author rebuttal - major comments**

We thank the reviewers for the valuable comments. We appreciate the understanding for the length of the paper, which was required to include important information.

We are happy to see that the reviewers consider the paper to be well written (R3), pinpoint our understanding of the subject and the depth of our work (R1, R3), and recognize our taxonomy to be a worthy effort (R2).

In the reviews, we see two main directions of criticism. We would like to disprove them here. We address other minor comments in additional individual replies under each review.

#1
The reviewers consider our paper to be a mere survey of existing methods, lacking novelty, and having low value for an experienced audience, thus unsuitable for ICLR.

We disagree with claims (mainly by R3) reducing our contribution to a mere review and classification of methods, because it is the classification scheme itself that is novel and central to our paper.

Our work is unprecedented in the scale of the analysis, considering methods ranging from regularization via data to studying the effects of optimization procedures. It is novel in the way it 1) decouples the "why" (assumptions the methods are enforcing) from the "how" (mathematical tools for the enforcement), 2) identifies "atomic" building blocks of the regularization methods, 3) simplifies discovery of new methods via recombinations of building blocks, and 4) offers a big picture by presenting relations between methods.

Researchers from applied fields may benefit from our taxonomy, as they can focus on the "why", i.e. discover new assumptions to be enforced. Thus, we believe our work has much higher application potential than just "a useful reference for students of deep learning" (R1) or "introductions in PhD thesis" (R3). On the other hand, deep learning researchers can focus on the "how" and discover new ways to enforce assumptions.

R1's claim that our suggestion to combine existing methods is "obvious" and "(the community) did not need a paper for that" is greatly undervaluing our contribution: We do not only say that combining of existing methods can yield new ones; we go further and identify atomic blocks, along with their benefits and limitations.

Moreover, we present novel perspectives on popular techniques (dropout from the optimization point of view, model compression in the data-transformation framework), contributing to their broader understanding.

We updated the abstract and introduction to better clarify these points. For the listed reasons, we kindly ask the reviewers to reconsider their ratings.

#2
R1 and R2 consider our definition of regularization to be too wide, encompassing also trainability (R2) and optimization methods (R1).

We believe it is the only correct approach. It is not so crucial what the optimum of the empirical risk is because 1) it cannot be found exactly, and 2) the empirical and expected risk are not equal; rather, the shape of the loss function and the optimization procedure play together to dictate how the training proceeds in the weight space and where it ends up; therefore, the effects of changing the loss function and optimization procedure are entangled and cannot be simply separated. The learned solution depends on all factors from Section 2.

This is supported by Zhang et al. (ICLR 2017), who demonstrate that explicit regularization is not sufficient to explain good generalization ability of deep nets. Also consider following examples of methods fitting the community understanding of "regularization" which can be simultaneously considered modifications to the optimization procedure or methods improving trainability:

- Dropout is considered regularization. In Section 7, Figure 1, we show how it can be interpreted as a modification to the optimization procedure.

- Weight decay is a regularizer, Krizhevsky et al. (2009) report it to help trainability of the network too.

- Narrowing down the initial hypothesis space is a form of regularization. Pre-training the network weights performs this implicitly (because it limits the subspace of weight configurations which the algorithm can in practice reach), thus it cannot be considered only trainability or optimization improvement.

- Batchnorm was designed to address trainability issues of deep nets; however, Ioffe and Szegedy (2015) also argue it works as a regularizer, introducing noise into the network through batch shuffling and reducing overfitting.

Such explanations (e.g. beginning of Section 4) are present in the paper; we also added a clarification to the beginning of Section 7 of the revised paper.

We hope that this demonstrates well that it is not possible to set a clear boundary between regularization and optimization/trainability; instead, they all must be considered when dealing with improving generalization of neural nets. Thus, we find this point of criticism invalid. We kindly ask the reviewers to reconsider their ratings.

Additional minor comments can be found under each review.

---

### Decision · Program_Chairs · 2018-01-29
**ICLR 2018 Conference Acceptance Decision**

**Decision:**

Reject

**Comment:**

The paper is a well-written review of regularization approaches in deep learning.
It does not offer novel approaches or novel insight with empirically demonstrated usefulness
=> ICLR is not the appropriate venue for it.